# Monitoring and Analysis of Prestress Loss in Prestressed Box Girder Bridges Strengthened with External Prestressing

**DOI:** 10.3390/s24144549

**Published:** 2024-07-13

**Authors:** Haoran Guo, Jing Yang, Renjie Tang, Caiqian Yang, Fu Xu

**Affiliations:** 1College of Civil Engineering, Xiangtan University, Xiangtan 411105, China; 202221572259@smail.xtu.edu.cn (H.G.); lingxuewei@njfu.edu.cn (R.T.); xufu@xtu.edu.cn (F.X.); 2College of Architectural Science and Engineering, Yangzhou University, Yangzhou 225127, China; 3School of Civil Engineering, Southeast University, Nanjing 210096, China; ycqjxx@seu.edu.cn

**Keywords:** prestress loss, external prestressing, monitoring, magnetic flux rope force transducers, Fiber Bragg Grating sensors

## Abstract

To investigate the effects of long-term prestress loss on concrete box girders strengthened with external prestressing, a large-span box girder, in service for over 20 years and strengthened with external prestressing, was monitored for four months. Prestress loss in the longitudinal, vertical, and transverse directions of the box girder was calculated according to Chinese code requirements. Magnetic flux rope force transducers were used to monitor the prestress loss in the external prestressing cables. Fiber Bragg Grating (FBG) sensors were used to monitor deflection changes at the mid-span of the bridge. Finally, the effect of prestress loss in the longitudinal, vertical, and transverse tendons on mid-span deflection was investigated through simulations using ABAQUS software. The results show that instantaneous prestress loss accounts for most of the total loss compared to long-term loss, and that longitudinal prestress loss has the most significant effect on mid-span deflection. The impact of longitudinal prestress loss on deflection before and after strengthening was also compared. The downward deflection and up-ward arch caused by longitudinal tendon prestress loss were reduced after strengthening, con-firming the effectiveness of the external prestressing method.

## 1. Introduction

With the urbanization of China, increasing traffic loads and environmental erosion have led to various performance degradation issues in traditional reinforced concrete bridge structures, such as steel reinforcement corrosion and concrete cracking within their design service life. Under the combined effects of fatigue loads and environmental factors, concrete girders are prone to cracking, resulting in decreased stiffness and reduced bearing capacity, significantly shortening the service life of bridges. Therefore, repairing, strengthening, and long-term monitoring of these structures are essential for preventing structural damage and improving structural design [1,2,3,4,5].

Various techniques can achieve structural strengthening, such as using flow able high-performance concrete (HPC) with higher ductility to increase shear load capacity by enlarging the end sections of hollow slab beams [6,7]. Additional strengthening materials, such as externally bonded fiber-reinforced polymers [8,9] or steel plates [10,11], can also be used to bond rectangular concrete beams, increasing their load-carrying capacity. However, although these methods have achieved good results, increasing the cross-section also increases the structural deadweight and leads to greater mid-span deflection. Additionally, using polymer or steel plates for adhesive reinforcement may affect the appearance and clear height of the bridge [7], making these methods more suitable for small and medium-span bridges rather than large-span bridges.

After being put into operation, large-span prestressed concrete (PC) continuous girders often exhibit continuous downward deflection that exceeds their expected design values and contradicts the theoretical analysis results [12], which suggest a constant value. To address this issue, researchers have adopted external prestressing reinforcement methods [13,14,15]. For instance, Park Y. H. [13] conducted field load tests and numerical simulations on a bridge operating for 23 years, revealing that external prestressing reinforcement effectively reduced structural downward deflection without significantly affecting the bridge’s intrinsic frequency and impact coefficient. Long-term monitoring showed that the external cable stress loss was negligible.

However, to improve stiffness and crack resistance, prestressing strands induce compressive stresses in the tensile portions of the members. Factors such as friction between prestressing strands and orifices [16], concrete shrinkage and creep [17], tendon stress relaxation [18], and deformation of anchorage devices can cause prestressing force loss along the member length during construction and tensioning, resulting in the effective prestressing force being lower than the tensioning control stress. Additionally, tension loss over time affects long-term deflection and service behavior. Therefore, obtaining data on prestress loss is crucial for assessing the safety and durability of prestressed members over time. Various methods have been studied, including vibrating-wire strain gauges (VWSG) [19], magnetic fluxes sensors [20], electromagnetic sensors [21,22], and Fiber Bragg Grating (FBG) sensors [23,24,25]. M. Lewis [19] obtained data on prestress loss by monitoring internal strain variations in the concrete, allowing the determination of the center of the prestressing reinforcement. However, this method only tracks prestressing losses due to concrete shrinkage and creep and cannot quantify losses due to stress relaxation. CHO et al. [21] used electromagnetic sensors to measure prestress distribution at the anchorage end of an actual girder and evaluate factors influencing the prestress. They derived a prestress calculation formula, but the method is limited to stress monitoring at the anchorage end. Abdel-Jaber [23] investigated strain variation at the centroid of cross-sectional stiffness (the centroid of the composite cross-section) using FBG sensors and validated the method through seven years of data collection on a post-tensioned concrete bridge at Princeton University. While many methods exist to monitor effective prestressing force, most of the above techniques are limited to laboratory measurements and general monitoring of prestressing force loss. In addition, there is a lack of comprehensive studies focusing on the long-term external prestress loss in large-span concrete girders, especially those strengthened with external prestressing.

This study aims to address the long-term external prestress loss in concrete girders strengthened with external prestressing, which has not been thoroughly investigated in the existing literature. To better understand the method described in this paper, Figure 1 illustrates the roadmap of the methodology. To achieve this, a strengthened box girder bridge was monitored for four months. First, the prestress losses in the longitudinal, vertical, and transverse directions of the box girder were calculated before strengthening, according to the Chinese code. Then, the external prestress loss and mid-span deflection increment were monitored over four months using magnetic flux rope force transducers and FBG sensors. Finally, the effects of longitudinal, vertical, and transverse prestressing tendon losses on mid-span deflection were investigated through simulations using ABAQUS software, showing that the effect of longitudinal prestressing loss was the most significant. The impact of longitudinal prestressing loss on mid-span deflection at different locations (top slab, mid-span bottom slab, and side-span bottom slab) was also investigated. The effects of different longitudinal tendon losses on deflection before and after strengthening were compared to verify the effectiveness of external prestressing.

## 2. Analysis of Long-Term Internal Prestress Loss

### 2.1. Project Background

The Tongyu River Bridge is located in Yancheng City, Jiangsu Province, China. The superstructure is a three-span, prestressed, continuous variable, cross-section box girder with spans of (45 + 80 + 45) m. The width of the top slab is 13.5 m, the width of the bottom slab is 6.0 m, and the cantilever length of the wing plate is 3.75 m. The height of the girder and the thickness of the bottom slab vary along a parabolic curve from the mid-span to positions away from the pier center. The girder height at the root is 4.5 m, and the thickness of the bottom slab is 1.2 m. At the center of the span, the girder height is 2.0 m, and the thickness of the bottom slab is 0.25 m. The detailed dimensions of the box girder are shown in Figure 2.

In Figure 3, the box girder of the main bridge uses a three-way prestressed system in the longitudinal, transverse, and vertical directions. The longitudinal prestressed tendons were low-relaxation 270-grade steel strands (Ey = 1.95 × 10^5^ MPa, $R_{y}^{b}$ = 1860 MPa), with an under-anchor control stress of 0.74 ${\;R}_{y}^{b}\;$ MPa. The prestressed strands for the top slab were OVM15-15, with a controlled tension of 2890.5 kN under a single-strand anchor. The bottom slab strands (mid-span bottom slab and side-span bottom slab) were OVM15-12, with a controlled tension of 2312.4 kN under a single-strand anchor. The transverse prestressed steel strands were Φ15-4, using low-relaxation 270-grade steel strands (Ey = 1.95 × 10^5^ MPa, $R_{y}^{b}$ = 1860 MPa). The strands were arranged along the direction of the bridge, with a basic spacing of 80 cm, and the alternating single-end tension anchorage method was adopted. The vertical prestressing steel bars were 32 mm diameter high-strength, fine-rolled threaded rebars with an ultimate strength of 750 MPa and a tensioning control tonnage of 542 kN.

### 2.2. Calculation of Long-Term Loss of Internal Prestressing

Numerous scholarly proposals have addressed the calculation of prestress loss. Notably, friction and anchorage losses vary with different tensioning and anchorage systems. Some researchers suggest that construction techniques can compensate for the effects of friction and anchorage losses. This article analyzes the method for calculating prestress loss based on the current Chinese highway bridge code [26], covering both instantaneous and long-term losses specified in the code.

Instantaneous loss mainly involves three aspects. First, straightening the prestressed tendons during tensioning is complex, and they have zero friction contact with the pipe wall. Since the actual tensile stress of the prestressed tendons is inevitably less than the tension force, considering the friction loss, expressed by the symbol $\sigma_{l1}$, is necessary. Based on Equation (1), where $\sigma_{con}$ is the tension control stress value under the anchorage of the prestressed steel, *μ* is the friction coefficient between the prestressed steel and the pipe wall, θ is the sum of the angles from the tension end to the tangent of the pipe section at the calculation section, *k* is the coefficient that influences friction due to local deviation per meter of the pipe, and *x* is the length of the pipe from the tension end to the calculation section.

Secondly, as the prestressing tendons are tensioned, the anchorage experiences significant pressure, causing deformation and shrinkage of the prestressed tendons. At the same time, the nodes of the assembled structure are also compressed. All these factors contribute to prestress loss, which needs to be considered. This includes prestressing loss caused by anchorage deformation, tendon shrinkage, and joint compression, expressed by the symbol $\sigma_{l2}$. According to Equation (2), ∆l is the total value of the anchorage deformation, tendon shrinkage, and joint compression, *l* is the distance from the tensioned end to the anchorage end, and $E_{p}$ is the elastic modulus of the prestressed tendons.

Finally, the structure undergoes elastic compression due to the action of prestressing. This deformation causes retraction of the prestressing tendons, leading to a loss of prestress. Therefore, it is necessary to consider the elastic compression loss of concrete, expressed by the symbol $\sigma_{l4}$*,* and calculated per Equation (3), where $∆\sigma_{pc}$ is the normal stress produced by the tensioning of the prestressing steel bars from the later batch. $\alpha_{EP}$ is the ratio of the elastic modulus of the prestressing tendons to the elastic modulus of concrete.
(1)$$\sigma_{l1}=\sigma_{con}\left[1-e^{-\left(\mu \theta +kx\right)}\right]$$
(2)$$\sigma_{l2}=\frac{\sum \Delta \sigma }{l}E_{P}$$
(3)$$\sigma_{l4}=\alpha_{EP}\sum \Delta \sigma_{pc}$$

Long-term losses mainly include two aspects, namely stress relaxation of prestressed reinforcement ($\sigma_{l5}$) and shrinkage and creep of concrete ($\sigma_{l6}$). Under long-term stress, the stress level of prestressed tendons decreases continuously over time, known as stress relaxation. It is calculated according to Equation (4), where Ψ is the tensioning coefficient, *ζ* is the relaxation factor of the steel, and $\sigma_{pe}$ is the prestressing tendon stress at the time of anchorage. On the other hand, over the long term, the shrinkage and creep of concrete also cause prestress loss. This aspect is calculated according to Equation (5). Where
$$\rho =\frac{A_{p}+A_{s}}{A}$$
$$\rho_{ps}=1+\frac{e_{ps}^{2}}{i^{2}}$$
(4)$$\sigma_{l5}=\psi \cdot \zeta \left(0.52\frac{\sigma_{pe}}{f_{pk}}-0.26\right)\sigma_{pe}$$
(5)$$\sigma_{l6}\left(t\right)=\frac{0.9\left[E_{p}\varepsilon_{cs}\left(t,t_{0}\right)+\alpha_{EP}\sigma_{pc}\varphi \left(t,t_{0}\right)\right]}{1+15\rho \rho_{ps}}$$

### 2.3. Results and Discussions

The above analysis shows that instantaneous losses are fully incurred upon the completion of the continuous box girder bridge. In contrast, long-term loss gradually increases over time. Table 1 and Table 2 record the changes in the true values and loss rates of the prestressed tendons in the top slab and web, respectively. Table 3 and Table 4 record the changes in the true values and loss rates in the prestressed tendons of the mid-span bottom slab and the side-span bottom slab, respectively. It can be observed that, compared to the initial controlled stress under anchor, after 15 years of bridge service, the losses of the top slab tendons ranged from 10% to 22%, the losses of the vertical tendons in the web ranged from 40% to 62%, the losses of the mid-span bottom tendons ranged from 20% to 24%, and the losses of the side-span bottom tendons ranged from 15% to 22%. These data reflect the extent of loss experienced by the prestressed tendons at different locations over the bridge’s entire service life. Notably, the loss of the top slab tendons is relatively small, while the loss of the vertical tendons in the web is the most significant.

Figure 4 shows the curves of prestress loss over time at different locations. From the figure, prestress loss can be divided into two time periods: before and after the completion of the bridge. Throughout the process, the transient prestress loss occurring at the completion of the bridge was dominant. Specifically, at the completion of the bridge, the top slab tendons experienced the largest loss of 21%. The loss range for web tendons was between 22% and 48%, and the mid-span and side-span bottom slab tendons had a maximum loss of 22%. Notably, the loss of prestress in the web tendons was higher at the completion of the bridge, likely due to their anchorage method. Compared to the prestressed tendons in the top and bottom slabs, the web tendons were anchored only at one end, resulting in weaker interactions between the tendons and the concrete [27].

Furthermore, one year after the completion of the bridge, the loss of prestress in the top and bottom slab tendons was approximately 1% to 2% compared to when the bridge was completed. The vertical prestressed tendons in the web exhibited relatively larger losses than other tendon groups, with losses ranging from 8% to 13%. After 15 years, the prestress in the top slab had decreased by a maximum of 1.1% (from 1095.3 MPa to 1083.0 MPa) compared to one year after completion. The loss of prestress in the web tendons was even more significant, ranging from 11.3% to 11.7% compared to one year after the bridge was completed. Additionally, the maximum loss of prestress in the mid-span and side-span bottom slab tendons after 15 years (331.2 MPa) had increased by 1.04 times compared to the maximum prestress one year after the bridge was completed (319.3 MPa). This indicates that the development of long-term prestress loss slows down noticeably over time.

## 3. Monitoring of Long-Term External Prestress Losses

### 3.1. Project Description

According to the inspection report, partial diagonal cracks were found in the web, along with mid-span deflection. The external prestressing strengthening method was used to reinforce the bridge and address these issues, with the prestressing loss monitored over time. During the monitoring process, force sensors were installed at the tensioning ends, and magnetic flux sensors were used to monitor the prestress loss of the external prestressed tendons. Measurements of prestress loss were taken in August, September, and November 2019 using vibrating string anchor cable gauges installed on the external prestressing tendons. A total of eight steel strands are arranged, with four strands on each side, namely T1 to T4, as shown in Figure 5c. The cross-section of section A is shown in Figure 5a, and the cross-section of section B is shown in Figure 5b.

### 3.2. Results and Discussions

The monitoring results of the external prestress loss are shown in Figure 6.

Deflection monitoring involved the deployment of distributed long-gauge FBG strain sensors to capture the structural strain response, which was subsequently combined with an improved conjugate beam method for calculating structural deflection. Long-term monitoring data from the inner side of the bridge span revealed that the left side inner beam deflected by 42 mm, while the right inner beam deflected by 36 mm. After strengthening, the difference between the initial wavelength of the longitudinal sensors in the stabilized stage before strengthening and the wavelength of the longitudinal sensors in the stabilized stage under no vehicular load during the operational stage was used to calculate the inverse arch value of the bridge. The bending moment-area method was applied to calculate the residual value of the inverse arch. Table 5 shows that the bridge’s arch reversal increased by 10.17 mm after strengthening. During the operational phase, the arch reversal values were 9.58 mm in July, 9.56 mm in August, 9.60 mm in September, and 9.56 mm in October, indicating that the overall deflection of the bridge was approximately 0.6 mm over a four-month operational period.

## 4. Simulation of the Effect of Prestress Loss on Deflection

A three-dimensional solid finite element model of a concrete box girder was established in ABAQUS to analyze the mechanical properties under varying prestress losses. The following simplifications were made: (1) piers were not simulated; (2) only half of the actual bridge was modeled due to the symmetric structure of the main bridge superstructure; (3) the influence of the slope was ignored; and (4) a linear connection was selected within each segment, neglecting the parabolic variation. The total length of the main bridge is 169.84 m. The C50 concrete was modeled using a concrete plastic damage model, and the mesh of the box girder bridge consisted of 8-node solid elements (C3D8R). The mechanical property parameters are as follows: a density of 2500 kg/m^3^, an elastic modulus of 3.45 × 10^4^ MPa, and a Poisson’s ratio of 0.2. External and longitudinal prestressed tendons were tensioned symmetrically from both ends. The vertical and transverse prestressed tendons were tensioned from a single end. The elastic modulus of the steel strand was 1.95 × 10^5^ MPa, the elastic modulus of the prestressed deformed steel was 2.00 × 10^5^ MPa, and the Poisson’s ratio was 0.3. The mesh consisted of T3D2 elements. The initial stress method was used to apply prestress to the steel strands, and the external prestressed tendons were constrained by embedded areas with anchor blocks and deviation blocks.

The materials mainly comprised reinforcement and concrete. The concrete damage plasticity (CDP) model characterizes the plastic mechanical behavior of concrete, simulating degradation effects from cracking and nonlinear deformation due to tension and compression [28]. As depicted in Figure 7a, the CDP model parameters for concrete material were set as follows [29]: expansion angle *φ* = 30°, eccentricity ratio *ε* = 0.1, initial biaxial to uniaxial compressive strength ratio *f_b0_/f_c0_* = 1.14, yield constant *K_c_* = 0.6667, and bond parameter *μ* = 0.0001. These parameters were selected to capture the complex behavior of concrete under tensile and compressive loads, ensuring the model accurately represents the mechanical response. The constitutive relationship of the reinforcement is shown in Figure 7b.

As shown in Figure 8, considering the structural symmetry, the constraints for the mid-span section (B1) in ABAQUS were set to XSYMM (U1 = UR2 = UR3 = 0). Under different prestress loss conditions, the constraints for the end section of the side-span (B2) restricted only the longitudinal (X-axis) translational degree of freedom to examine the deformation. Fixed boundary conditions were applied to the bottom support of the box girder (B3), restricting displacements U1, U2, and U3, as well as rotations UR1, UR2, and UR3. The constraints at the bottom support of the side-span (B4) restricted only the vertical displacement (U2 = 0). The loads were divided into two components: the total bridge weight and the deck pavement. In the actual bridge, the deck pavement consists of 10 cm thick asphalt concrete. In the model, this is converted into pressure applied to the bridge.

### 4.1. Loss of Prestress in Three Directions

The previous analysis shows that prestress loss is influenced by factors such as the length and curvature of the tendons, as well as concrete shrinkage and creep. Differences in the size and positions of the three-directional prestressed tendons result in variations in prestress loss. Specifically, the longitudinal prestress loss is about 20%, the maximum vertical prestress loss reaches 61.97%, and the transverse prestress loss is approximately 50%. These data reveal significant differences in prestress loss in different directions, providing a basis for subsequent research. Therefore, to comprehensively explore the influence of internal and external prestress loss on mid-span deflection before and after strengthening the box girder bridge, the longitudinal prestress loss was assumed to be 0%, 10%, 20%, and 30%. Meanwhile, the effect of transverse and vertical prestressing was considered, with the degree of loss set at 0%, 20%, 40%, and 60%. Additionally, the external prestress loss was set at 0%, 10%, 20%, and 30% to analyze the influence of prestress loss on structural performance. The specific results are shown in Figure 9, Figure 10, Figure 11, Figure 12 and Figure 13.

#### 4.1.1. Loss of Longitudinal Prestress

Figure 9a shows the deflection of the box girder bridge before strengthening, under longitudinal prestress losses of 0%, 10%, 20%, and 30%. The deflections under the bottom slab at mid-span are 12.96 mm, 27.63 mm, 42.11 mm, and 52.24 mm, respectively. These data reveal a direct correlation: as longitudinal prestress loss increases, the mid-span deflection of the bridge also increases. Figure 9b displays the deflection difference between prestress losses of 10%, 20%, and 30% compared to 0%. As prestress loss increases, the deflection difference grows progressively from 14.67 mm to 39.28 mm. This shows that the increase in longitudinal prestress loss has a significant impact on structural deflection.

Figure 9c shows the deflection of the box girder bridge after external prestressing strengthening for longitudinal prestress losses of 0%, 10%, 20%, and 30%. The deflections under the bottom slab at mid-span are 11.04 mm, 21.76 mm, 32.37 mm, and 38.60 mm. Compared to before strengthening, there is a considerable decrease in deflection. Figure 9d shows the deflection difference curve, indicating that as prestress loss increases, the difference in deflection increases from 10.72 mm to 27.56 mm. Specifically, when the prestress loss is 0%, the deflection decreases from 12.96 mm before strengthening to 11.04 mm after strengthening, representing a reduction of 14.8%. This indicates that structural stiffness increases after external prestressing strengthening. For prestress losses of 10%, 20%, and 30%, the deflection reductions are 21.2%, 23.1%, and 26.1%, respectively. It can be observed that the deflection reduction rate increases gradually with the increase in longitudinal prestress loss. This indicates that the larger the longitudinal prestress loss, the more significant the effect of external prestressing strengthening on reducing mid-span deflection.

**Figure 9 sensors-24-04549-f009:**
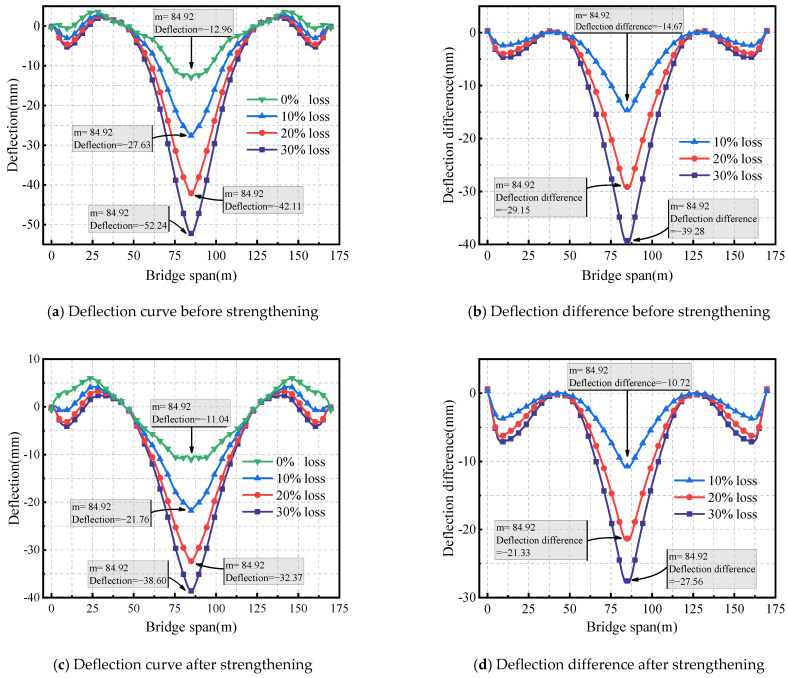
Effect of longitudinal prestressed tendons loss on deflection.

#### 4.1.2. Loss of Transverse Prestress

Figure 10a shows the deflection curve of the box girder bridge before strengthening, with transverse prestress losses of 0%, 20%, 40%, and 60%. For transverse prestress losses of 0%, 20%, 40%, and 60%, the deflections at the mid-span of the bridge are 12.96 mm, 12.53 mm, 11.94 mm, and 11.17 mm, respectively. This indicates that transverse prestress loss leads to an intermediate reverse arch at mid-span. As shown in Figure 10b, when the transverse prestress loss reaches 60%, the reverse arch is 1.79 mm. Additionally, transverse prestress loss results in a downward deflection at the mid-span of the side-span. When the transverse prestress loss reaches 60%, the maximum deflection difference at the mid-span of the side-span is 0.14 mm.

Figure 10c shows the deflection of the box girder bridge after external prestressing strengthening for transverse prestress losses of 0%, 20%, 40%, and 60%. For transverse prestress losses of 0%, 20%, 40%, and 60%, the deflections at the mid-span of the bridge are 11.04 mm, 10.80 mm, 10.37 mm, and 9.88 mm, respectively. Figure 10d shows the deflection difference curve. Similar to before strengthening, transverse prestress loss leads to an upward deflection at mid-span after strengthening. When the transverse prestress loss reaches 60%, the mid-span upward deflection is 1.16 mm. However, the loss of transverse prestress also causes an upward deflection at the mid-span of the side-span. When the transverse prestress loss reaches 60%, the maximum deflection difference is 0.28 mm. It can be observed that the amplitude of the mid-span upward deflection is reduced after strengthening, but the amplitude of the mid-span upward deflection in the side-span is increased. When the prestress loss is 0%, the deflection decreased from 12.96 mm before strengthening to 11.04 mm after strengthening, representing a reduction of 14.8%. This suggests an increase in stiffness after strengthening. Despite a transverse prestress loss of 30% after strengthening, the deflection decreases by 11.5%. This further confirms the effectiveness of the strengthening, as it still maintains good performance improvement even with a certain level of transverse prestress loss.

**Figure 10 sensors-24-04549-f010:**
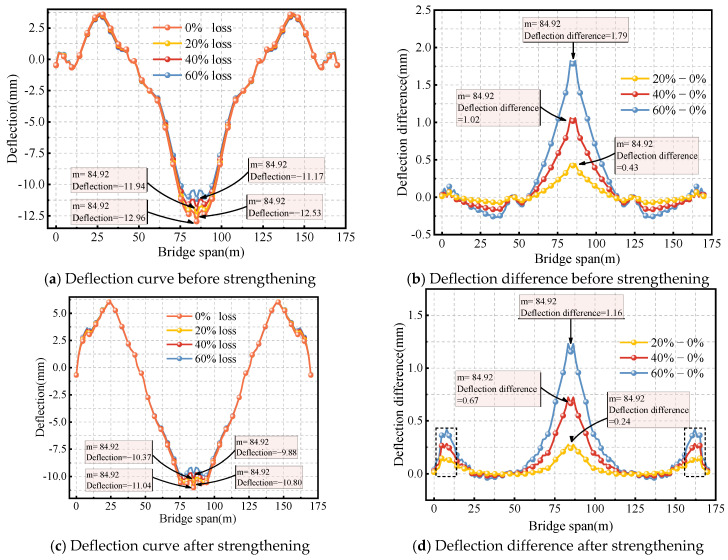
Effect of transverse prestressed tendons loss on deflection.

#### 4.1.3. Loss of Vertical Prestress

Figure 11a shows the deflection of the box girder bridge before strengthening when the vertical prestress loss is 0%, 20%, 40%, and 60%. The deflections at the mid-span bottom slab are 12.96 mm, 12.97 mm, 13.17 mm, and 13.50 mm, respectively. This indicates that the mid-span deflection increases gradually as the vertical prestress loss increases. As depicted in Figure 11b, compared to the situation without loss, the deflection increases by 0.54 mm when the vertical prestress loss reaches 60%. Additionally, the loss of vertical prestress results in slight arching at the side-span. When the vertical prestress loss reaches 60%, the maximum difference in arching at the side-span is 0.26 mm.

Figure 11c demonstrates that when the vertical prestress loss is 0%, 20%, 40%, and 60%, the mid-span of the bridge deflects downward by 11.04 mm, 11.18 mm, 11.36 mm, and 11.53 mm, respectively. In Figure 11d, the deflection difference curve is similar to before strengthening. After strengthening, the loss of vertical prestress leads to an increase in deflection at mid-span and arching at the side-span. When the vertical prestress loss reaches 60%, the arching at the side-span measures 0.25 mm. When the prestress loss is 0%, the deflection decreases from 12.96 mm before strengthening to 11.04 mm after strengthening. The deflection maintains a significant reduction of 14.6% even with a 60% loss of vertical prestress.

**Figure 11 sensors-24-04549-f011:**
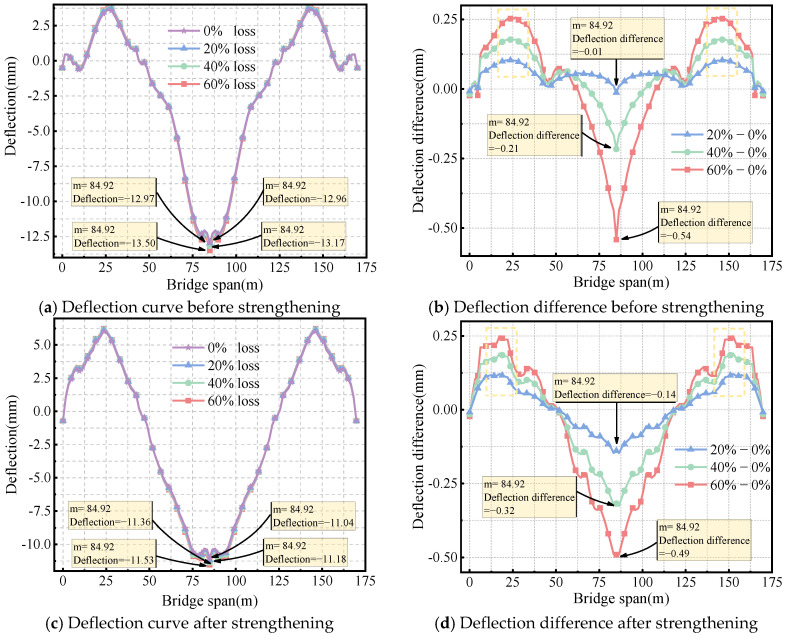
Effect of vertical prestressed tendons loss on deflection.

### 4.2. Loss of Longitudinal Prestress at Different Locations

The previous analysis clearly shows that the loss of longitudinal prestress has the most significant impact on deflection. To further explore this influence, longitudinal prestressed tendons were selected at different locations (top slab, mid-span bottom slab, and side-span bottom slab). Various levels of prestress loss, ranging from 0% to 30%, were assumed and analyzed for their effects on deflection.

#### Before Strengthening

Figure 12a–c show the deflection curves for longitudinal prestressed tendons at different locations (top slab, mid-span bottom slab, and side-span bottom slab) with prestress losses of 0%, 10%, 20%, and 30% before strengthening. Figure 12 shows differences in the effect of longitudinal prestressed tendons on deflections due to their different locations in the bridge. When the prestressed tendons at the top slab, mid-span bottom slab, and side-span bottom slab experienced a 10% loss, the deflections at mid-span were 30.75 mm, 8.86 mm, and 12.74 mm, respectively. With a 30% prestress loss, the deflections at mid-span were 54.94 mm, 12.96 mm, and 13.07 mm, respectively. It can be observed that prestress loss at the top slab has the greatest influence on mid-span deflection, followed by the side-span bottom slab and then the mid-span bottom slab. Figure 12d–f show the corresponding deflection difference plots. It can be seen that as the prestress loss at the top slab increases, the rate of deflection increase gradually decreases.

With a 10% prestress loss in the mid-span bottom slab, the side-span deflects 1.02 mm, while the mid-span arch is 4.10 mm. However, with a 30% loss of mid-span bottom slab tendons, the side-span deflection is 2.82 mm, while the mid-span upper arch is 12.40 mm. This indicates that the loss of prestress in the mid-span bottom slab leads to upward arching at the mid-span and downward deflection at the side-span. Finally, prestress loss in the bottom slab of the side-span causes downward deflection of the side-span but has less effect on the mid-span. When there is a 30% prestress loss in the side-span bottom slab, the side-span will deflect 3.19 mm.

Figure 13 illustrates the deflection curves of longitudinal prestressing tendons at different locations under varying levels of loss after strengthening. Similar to the situation before strengthening, it can be seen from Figure 13d that the loss of external prestressing has a relatively small effect on deflection. In contrast, the impact of prestress loss at different positions is more significant on the deflection of the side-span than on the mid-span. Specifically, when the prestress loss is 10%, the deflections of the side-span and mid-span are 5.43 mm and −10.94 mm, respectively; when the prestress loss is 30%, the deflections are 4.28 mm and −10.44 mm, respectively. The mid-span deflection of the side-span increased by 21.2%, while the reverse arch of the mid-span increased by 4.57%. It can be observed that the loss of prestressing tendons causes an increase in the side-span deflection, while the mid-span experiences a slight reverse arch. This indicates that external prestressing strengthening effectively enhances the stiffness of the bridge, achieving a satisfactory reinforcement effect.

**Figure 12 sensors-24-04549-f012:**
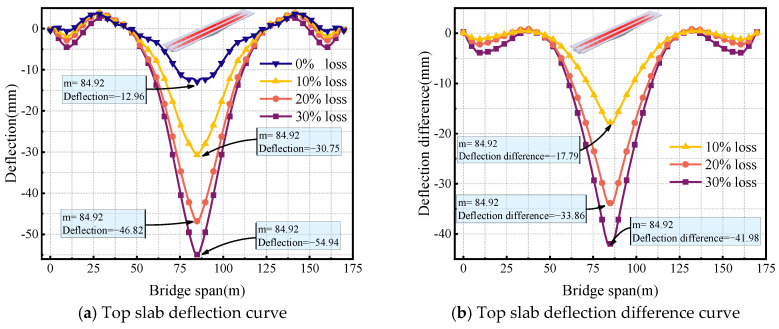
Effect of longitudinal prestress loss at different locations on deflection before strengthening.

**Figure 13 sensors-24-04549-f013:**
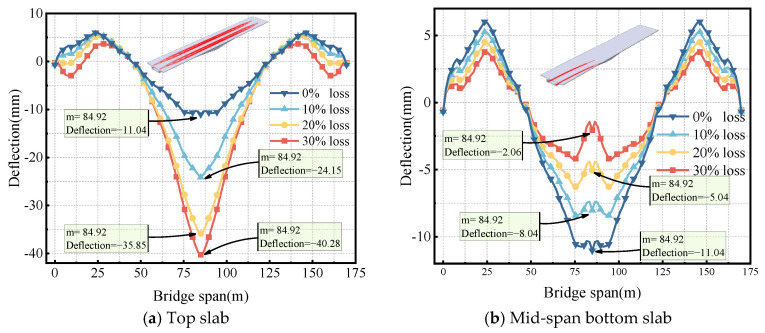
Effect of longitudinal prestress loss at different locations on deflection after strengthening.

## 5. Implications and Feasibility

### 5.1. Implications

The findings of this study have significant implications for the field of bridge engineering, particularly in the monitoring and maintenance of prestressed concrete girders. By distinguishing between instantaneous and long-term prestress losses, this research provides a clearer understanding of the critical periods for monitoring and intervention. The detailed analysis of how longitudinal, transverse, and vertical prestressing losses affect bridge deflection offers valuable insights for optimizing the design and maintenance strategies of prestressed bridges.

1. Enhanced Monitoring Practices: The dominant role of instantaneous prestress loss suggests that monitoring efforts should be intensified during and immediately after the construction phase. This can help in the early detection and mitigation of significant prestress losses, thereby ensuring the structural integrity and safety of the bridge.

2. Design Optimization: The study highlights the critical role of longitudinal prestressing in controlling mid-span deflection. These insights can inform the design of new bridges and the retrofitting of existing ones to enhance overall performance.

3. External Prestressing Reinforcement: The effectiveness of external prestressing reinforcement in reducing deflections is evident from the study. This technique can be widely adopted in both new constructions and rehabilitation projects to enhance the durability and load-carrying capacity of bridges.

### 5.2. Feasibility

The feasibility of implementing the monitoring and reinforcement techniques proposed in this study is high, given the advancements in sensor technology and engineering practices.

1. Sensor Technology: The use of magnetic flux rope force transducers and Fiber Bragg Grating (FBG) sensors has proven effective in accurately monitoring prestress losses and deflection changes. These sensors are readily available and can be integrated into existing bridge monitoring systems with minimal modifications.

2. Practical Application: The methods described for monitoring and reinforcing prestressed concrete girders are practical and have been validated through field studies and simulations. The techniques are not only feasible but also scalable, allowing for application in a wide range of bridge sizes and types.

3. Cost-Effectiveness: While initial setup costs for advanced monitoring systems and external prestressing reinforcement may be high, the long-term benefits of increased bridge lifespan, reduced maintenance costs, and enhanced safety outweigh the initial investment. The cost-effectiveness of these methods makes them a viable option for bridge management authorities.

4. Implementation Challenges: Potential challenges in implementing these techniques include the need for skilled personnel to install and maintain the monitoring equipment and the requirement for occasional calibration and validation of sensors. However, these challenges can be addressed through proper training and the use of automated systems for calibration and data analysis.

## 6. Conclusions

This study provides crucial insights into prestress loss, highlighting the need for monitoring to maintain bridge integrity and safety. Engineers should focus on longitudinal prestress loss, especially in the top slab, and consider external prestressing reinforcement for effective bridge rehabilitation. Regular monitoring and timely interventions are essential for extending the service life of prestressed concrete bridges. By addressing prestress loss comprehensively, this study offers valuable guidance for future bridge maintenance and engineering efforts. The key findings from each section of the manuscript are summarized as follows:Prestress loss can be divided into instantaneous loss and long-term loss. Before the bridge was completed, the instantaneous loss of prestressing tendons was predominant. In this phase, the prestress loss in the top slab was relatively small, ranging from 10% to 22%, whereas the vertical prestress loss in the web was the most significant, reaching 40% to 62%. After the bridge was completed, the long-term loss was relatively minor, and the progression of long-term prestress loss significantly slowed down over time.Magnetic flux rope force transducers were used to monitor prestress loss in external prestressing tendons, while FBG sensors were employed to measure the deflection changes at the mid-span of the bridge.Through simulations conducted using ABAQUS software, the study explored the effects of internal and external prestress loss on mid-span deflection before and after the strengthening of the box girder bridge. The results indicated that longitudinal prestress loss had the most significant impact on mid-span deflection. Without external prestressing reinforcement, when the longitudinal prestress loss increased from 0% to 30%, the mid-span bottom slab deflection increased from 12.96 mm to 52.24 mm. Transverse prestress loss caused reverse arching at mid-span; when transverse prestress loss reached 60%, the reverse arching was 1.79 mm. Vertical prestress loss led to reverse arching in the side spans and downward deflection at mid-span. As vertical prestress loss increased, the mid-span deflection gradually increased.The study further examined the impact of longitudinal prestress loss in the top slab, web, and bottom slab on mid-span deflection. The results showed that longitudinal prestress loss in the top slab had the most significant impact on mid-span deflection, followed by the bottom slab of the side spans, and lastly, the mid-span bottom slab. Longitudinal prestress loss in the mid-span bottom slab caused upward arching at mid-span and downward deflection in the side spans. Prestress loss in the side-span bottom slab primarily caused downward deflection in the side spans, with minimal impact on the mid-span.Comparing the effects of longitudinal, transverse, and vertical prestress losses on deflection before and after strengthening, it was found that the downward deflection caused by internal prestress loss decreased after strengthening. This indicates that external prestressing reinforcement effectively improved the stiffness of the bridge, achieving the desired strengthening effect.

## Figures and Tables

**Figure 1 sensors-24-04549-f001:**
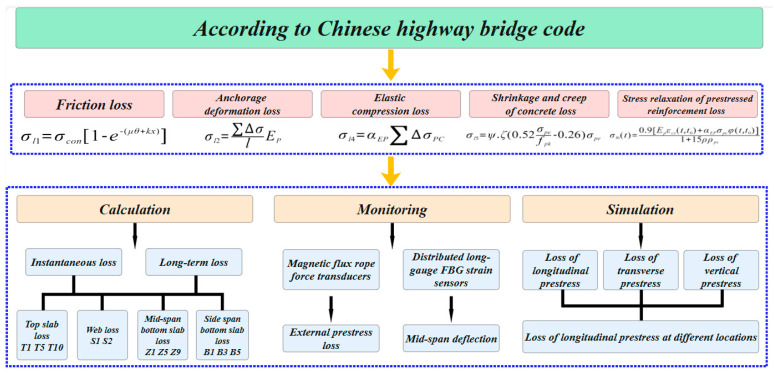
The roadmap of the methodology.

**Figure 2 sensors-24-04549-f002:**
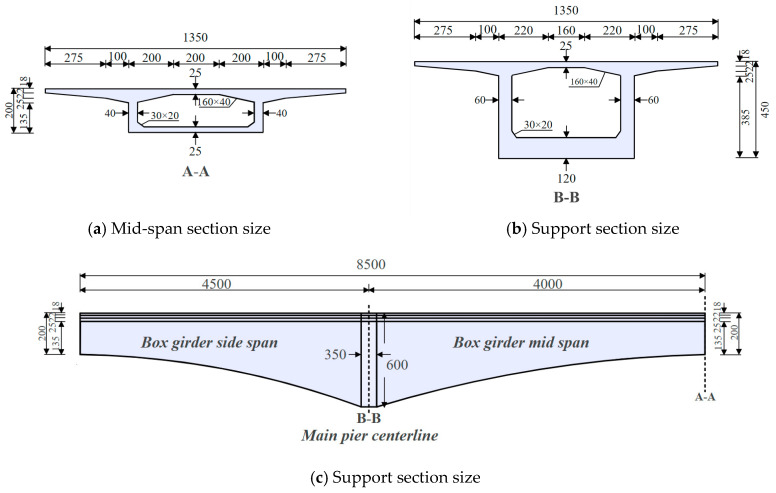
Elevation and cross-sectional dimensions (unit: cm).

**Figure 3 sensors-24-04549-f003:**
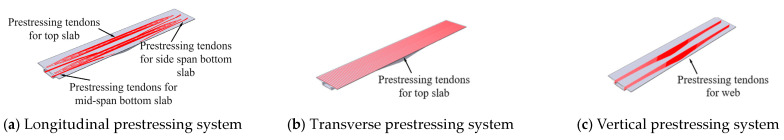
Three-way prestressing system.

**Figure 4 sensors-24-04549-f004:**
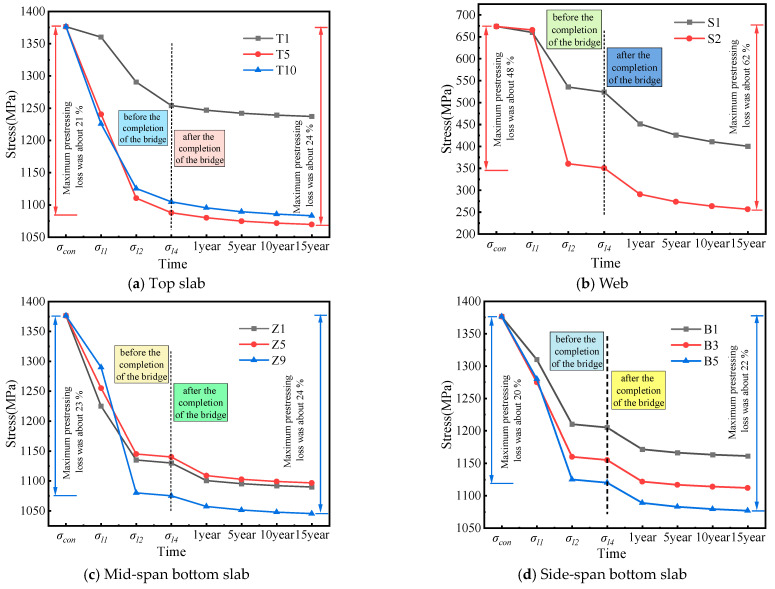
Prestress loss as time curve.

**Figure 5 sensors-24-04549-f005:**
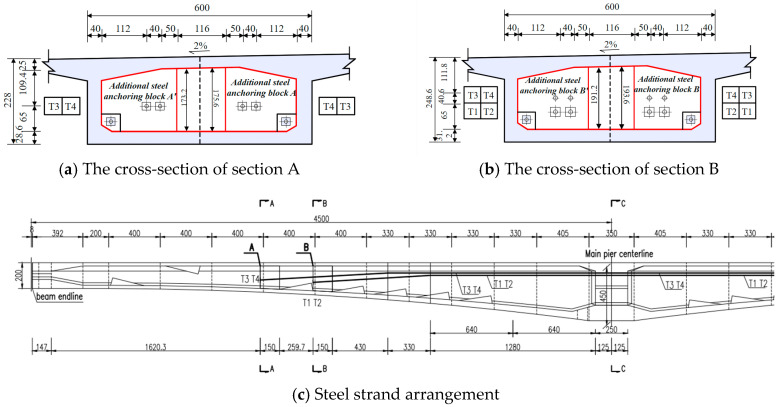
Positions of the external prestressing tendons.

**Figure 6 sensors-24-04549-f006:**
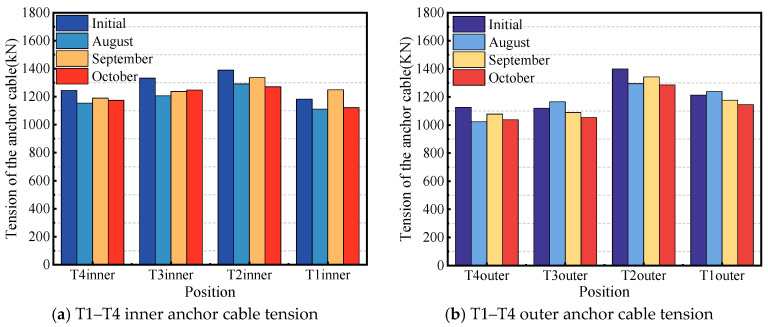
External prestressing loss monitoring.

**Figure 7 sensors-24-04549-f007:**
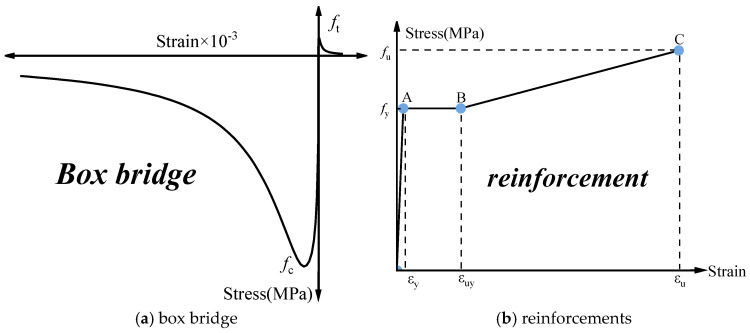
Constitutive stress–strain laws for box bridge and reinforcements.

**Figure 8 sensors-24-04549-f008:**
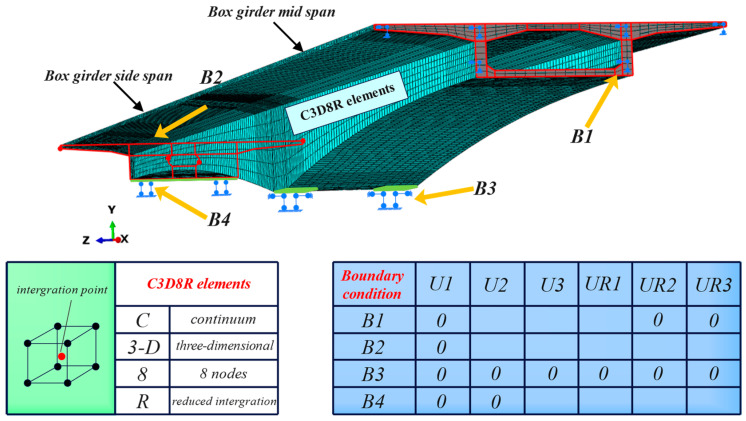
General details of the finite element modeling.

**Table 1 sensors-24-04549-t001:** Prestress loss results for top slab and web tendons over time (unit: MPa).

Reference Number	Top Slab			Web	
	T1	T5	T10	S1	S2
	(True Value)				
Controlled stress under anchor	1376.4	1376.4	1376.4	673.9	673.9
At the time of completion	1254.0	1087.6	1104.5	523.8	350.3
After 1 year	1246.8	1079.8	1095.3	451.1	290.5
After 5 years	1242.0	1074.6	1089.2	425.7	273.4
After 10 years	1239.1	1071.5	1085.5	410.5	263.3
After 15 years	1237.1	1069.4	1083.0	400.1	256.3

**Table 2 sensors-24-04549-t002:** Prestress loss ratio results for top slab and web tendons over time.

Reference Number	Top Slab			Web	
	T1	T5	T10	S1	S2
	(Loss Ratio%)				
Controlled stress under anchor	/	/	/	/	/
At the time of completion	8.89	20.98	19.75	22.28	48.02
After 1 year	9.42	21.55	20.42	33.06	56.89
After 5 years	9.77	21.93	20.87	36.84	59.43
After 10 years	9.98	22.15	21.14	39.08	60.93
After 15 years	10.12	22.31	21.32	40.63	61.97

**Table 3 sensors-24-04549-t003:** Prestress loss calculation results for bottom slab tendons over time (unit: MPa).

Reference Number	Mid-Span Bottom Slab			Side-Span Bottom Slab		
	Z1	Z5	Z9	B1	B3	B5
	(True Value)					
Controlled stress under anchor	1376.4	1376.4	1376.4	1376.4	1376.4	1376.4
At the time of completion	1108.7	1117.9	1066.1	1179.2	1129.1	1097.7
After 1 year	1100.6	1108.8	1057.1	1171.5	1121.7	1088.7
After 5 years	1095.2	1102.7	1051.2	1166.4	1116.8	1082.8
After 10 years	1092.0	1099.1	1047.6	1163.3	1113.9	1079.2
After 15 years	1089.7	1096.6	1045.2	1161.2	1111.9	1076.7

**Table 4 sensors-24-04549-t004:** Prestress loss ratio calculation results for bottom slab tendons over time.

Reference Number	Mid-Span Bottom Slab			Side-Span Bottom Slab		
	Z1	Z5	Z9	B1	B3	B5
	(Loss Ratio%)					
Controlled stress under anchor	/	/	/	/	/	/
At the time of completion	19.45	18.78	22.55	14.33	17.97	20.25
After 1 year	20.04	19.44	23.20	14.89	18.50	20.90
After 5 years	20.43	19.88	23.63	15.26	18.86	21.33
After 10 years	20.66	20.15	23.89	15.48	19.07	21.59
After 15 years	20.83	20.33	24.07	15.63	19.22	21.77

**Table 5 sensors-24-04549-t005:** Bridge mid-span reverse arch values from July to November.

Month	Mid-Span Reverse Arch Value (mm)
After tensioning	10.17
July	9.58
August	9.56
September	9.60
October	9.56

## Data Availability

The original contributions presented in the study are included in the article, further inquiries can be directed to the corresponding authors.

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
