# Peer review of "Monitoring and Analysis of Prestress Loss in Prestressed Box Girder Bridges Strengthened with External Prestressing"

_sensors, 2024, doi:10.3390/s24144549_

Round 1
Reviewer 1 Report
Comments and Suggestions for Authors
The paper investigates monitoring and analysis of prestress loss in prestressed box girder bridges strengthened with external prestressing. Some comments which greatly enhance the understanding of the paper and its value are presented below. Specific issues that require further consideration are:
- The methodology section is not clear, you may summarize it in a figure or chart.
- In lines 49 and 135, use the terminology of “Researchers” instead of “Scholars”.
- In lines 110 to 123, the symbols are off the lines.
- Improve the style of Tables 1 and 2.
- Figure 4 is vague, add caption for each picture and add more details to describe these pictures in section 3.1.
- Figure 6 is not professional. Also, you should show the boundary conditions and loading definition. Did you conduct the sensitivity analysis to find the optimum mesh size? Show the stress-strain diagrams of the concrete and steel that you used in the FEM model. How did you define the prestressing forces in the ducts in the FEM model?
- You should show the deformed shape (vertical deflection) of the FEM model for the results that you mentioned in Figures 7 and 8.
- Figures 10 and 11 are not clear (low resolution) and are not readable.
- You should show the deformed (Mises stresses) FEM model for the results that you mentioned in Figures 10 and 11.
10. The implications and feasibility of this work should be discussed in a new section before the Conclusions section.
11. The conclusions section should be more comprehensive summarizing key information of all sections in this manuscript.
Comments on the Quality of English LanguageExtensive editing of English language required
Reviewer 2 Report
Comments and Suggestions for Authors
Dear Authors,
The paper is interesting, but a few corrections are necessary.
1. Figures must be positioned next to where they are mentioned in the text. For example, figure 1 should be inserted from line 100.
2. Font size must be the same throughout the whole paper (see lines 105 and 106 compared to lines 107 and 108).
3. Certain details on the figures must be written with a larger font, close to the font size of the paper’s text. For example, see figure 4.
4. Editing the article must be done in such a way that the figures fit entirely on a single page (see figure 5).
5. The quality of some figures needs to be improved. Font size of the figures must be comparable to that of the paper’s text and the same for all figures. A higher resolution of figures (minimum 600 dpi) is required. See figures 10, 11!!!!
6. Check the correct way of numbering the equations in the article.
7. English is good. Check for missing spaces, commas or periods or spelling mistakes.
Comments on the Quality of English LanguageEnglish is good. Check for missing spaces, commas or periods or spelling mistakes.
Reviewer 3 Report
Comments and Suggestions for Authors
Review of manuscript Sensors-3082568
By Haoran Guo et al.
Submitted to Sensors
Monitoring and analysis of prestress loss in prestressed box girder bridges strengthened with external prestressing
General Comments:
The manuscript "Monitoring and Analysis of Prestress Loss in Prestressed Box Girder Bridges Strengthened with External Prestressing" presents a thorough investigation into the effects of long-term prestress loss on concrete box girders strengthened with external prestressing. The study is well-motivated, addressing the critical issue of maintaining structural integrity and longevity of large-span bridges. The methodology is comprehensive, involving both experimental monitoring and numerical simulations.
While the study addresses important aspects of the topic, there are critical areas that need improvement to meet the rigorous standards of "Sensors". The current version CANNOT be accepted in its present form due to the following issues. Furthermore, suggestions and comments are provided to enhance the manuscript's readability and overall quality.
Major/Minor Comments:
l The introduction effectively sets the context for the study, highlighting the importance of addressing prestress loss in large-span bridges. The literature review is thorough, but it would benefit from a clearer identification of the research gap. The motivation for the study should be more explicitly linked to the existing literature. The objectives of the study are stated but could be more succinctly defined.
l The description of the monitoring setup using magnetic flux rope force transducers and Fiber Bragg Grating (FBG) sensors is detailed. However, the rationale for selecting these specific sensors over others mentioned in the literature should be discussed.
l The methodology for data collection over four months is appropriate. It would be beneficial to provide more details on the calibration of sensors and the handling of potential data anomalies.
l The use of ABAQUS software for simulations is suitable. The manuscript should include more detailed information on the boundary conditions and the assumptions made during the simulation process.
l The results are presented logically, showing the relationship between prestress loss and mid-span deflection. The figures and tables are clear but could be better integrated into the text. Each figure and table should be discussed in more detail within the narrative to enhance understanding. The manuscript should provide more quantitative comparisons, such as percentage reductions in deflection due to external prestressing.
l The discussion interprets the results in the context of existing knowledge. However, it could be expanded to explore the implications of the findings more deeply. The manuscript should discuss potential limitations of the study, such as the four-month monitoring period, and suggest directions for future research.
l The conclusion summarizes the findings but is somewhat brief. It should restate the main contributions of the study and their significance in the broader context of bridge engineering. Practical implications and recommendations for engineers should be included.
Comments on the Quality of English LanguageModerate editing of English language required
Round 2
Reviewer 1 Report
Comments and Suggestions for Authors
1. The response of comment #1 should be reflected on the manuscript to describe the methodology of this study.
2. More details regarding Figure 6 are needed to describe the content, also, what are the references of the steel and concrete models?
Comments on the Quality of English Language
Minor editing of English language required
Reviewer 2 Report
Comments and Suggestions for Authors
Dear Authors,
Some improvements have been made, but a few issues reported in review 1 still remain.
1. In Figure 3, the text on the right side of the 4 representations a) - d) must be rotated by 180 degrees. Notations written on the abscissa of the graphs must be written correctly, using the "Subscript" option where necessary. An example, σl1 should be written σl1.
2. Edit representation a) in Figure 2 so the text on it is similar to the texts on representations b) and c).
3. Notations written in the text of the article must be written correctly, using the "Subscript" option (see σl1 – line 137, σl2 – line 149, etc.).
Comments on the Quality of English LanguageEnglish is good. Check for missing spaces, commas or periods or spelling mistakes.
Reviewer 3 Report
Comments and Suggestions for Authors
After a thorough review of the revisions, I am pleased to report that the authors have made significant improvements addressing the concerns and suggestions raised during the first review. The revisions have enhanced the clarity, rigor, and overall quality of the manuscript.
Given these improvements, I am pleased to recommend the manuscript for acceptance. I believe it will be a valuable contribution to the field and will be well-received by the journal's readership.
Comments on the Quality of English LanguageMinor editing of English language required
